# Genetic-Based Hypertension Subtype Identification Using Informative SNPs

**DOI:** 10.3390/genes11111265

**Published:** 2020-10-27

**Authors:** Yuanjing Ma, Hongmei Jiang, Sanjiv J Shah, Donna Arnett, Marguerite R Irvin, Yuan Luo

**Affiliations:** 1Department of Statistics, Northwestern University, Evanston, IL 60208, USA; yuanjingma2020@u.northwestern.edu (Y.M.); hongmei@northwestern.edu (H.J.); 2Feinberg School of Medicine, Northwestern University, Chicago, IL 60611, USA; sanjiv.shah@northwestern.edu; 3College of Medicine, University of Kentucky, Lexington, KY 40506, USA; donna.arnett@uky.edu; 4School of Public Health, University of Alabama at Birmingham, Birmingham, AL 35294, USA; irvinr@uab.edu

**Keywords:** hypertension, variable selection, subtype identification, clustering algorithm, NMF

## Abstract

In this work, we proposed a process to select informative genetic variants for identifying clinically meaningful subtypes of hypertensive patients. We studied 575 African American (AA) and 612 Caucasian hypertensive participants enrolled in the Hypertension Genetic Epidemiology Network (HyperGEN) study and analyzed each race-based group separately. All study participants underwent GWAS (Genome-Wide Association Studies) and echocardiography. We applied a variety of statistical methods and filtering criteria, including generalized linear models, F statistics, burden tests, deleterious variant filtering, and others to select the most informative hypertension-related genetic variants. We performed an unsupervised learning algorithm non-negative matrix factorization (NMF) to identify hypertension subtypes with similar genetic characteristics. Kruskal–Wallis tests were used to demonstrate the clinical meaningfulness of genetic-based hypertension subtypes. Two subgroups were identified for both African American and Caucasian HyperGEN participants. In both AAs and Caucasians, indices of cardiac mechanics differed significantly by hypertension subtypes. African Americans tend to have more genetic variants compared to Caucasians; therefore, using genetic information to distinguish the disease subtypes for this group of people is relatively challenging, but we were able to identify two subtypes whose cardiac mechanics have statistically different distributions using the proposed process. The research gives a promising direction in using statistical methods to select genetic information and identify subgroups of diseases, which may inform the development and trial of novel targeted therapies.

## 1. Introduction

Hypertension is a heterogeneous condition with various subtypes that differ in pathophysiology and age distribution [1]. It has been shown to have subtypes such as borderline isolated systolic hypertension (systolic ≥140, diastolic <90 mmHg), definite isolated systolic hypertension (systolic ≥160, diastolic <90 mmHg), isolated diastolic hypertension (systolic <160, diastolic ≥90 mmHg), and definite hypertension (systolic ≥160 and diastolic ≥90 mmHg or hypertensives on treatment) [2]. The discovery of hypertension phenotypic heterogeneity leads us to pursue the next question of identifying subtypes of hypertension based on genetic information.

Development of high throughput technologies, such as genome-wide sequencing and SNP-array approaches give researchers promising directions to reveal disease genes and mechanisms for further application to new therapeutic targets. To better achieve effectiveness, it is important to select informative genetic information for down-stream analysis such as clustering and classification. Different machine learning and statistical learning algorithms have been applied to select useful information and group together patients with similar phenotypic and genetic performance. Supervised learning can be applied to find out disease related variants. The support vector machine was used to determine the SNP subset which has the highest classification accuracy [3]. Sequence kernel association test (SKAT) was developed to find out association between the genetic region and a continuous or dichotomous trait while adjusting for covariates [4]. Unsupervised learning, on the other hand, is to learn the intrinsic structures of selected informative genetic data to derive novel subtypes of disease. Brunet, Tamayo et al. applied non-negative matrix factorization to solve three problems in elucidating cancer subtypes [5]. Cai, Huang et al. modified K-means clustering in analysis of SAGE data [6]. Shah, Katz et al. used unbiased hierarchical cluster analysis of phenotypic and penalized model-based clustering, to define and characterize mutually exclusive groups making up a novel classification of heart failure with preserved ejection fraction [7].

In this paper, we use a generalized linear model to limit the scope of variants to hypertension related ones and then apply different criteria to further filter informative SNPs to reduce the false positive rate. In addition to common genetic filters such as rare variants and deleterious variants, we also make a generalization of F-ranking statistics. F-statistics describe the level of a variant’s variability among different population classes. Variants with higher F statistics have more capability in distinguishing diseases status. Therefore, we can use F statistics as a metrics to measure the relatedness of a variant with hypertension status. For the choice of unsupervised learning algorithm, various methods have been developed to cluster samples with similar genetic information. In this paper, we use non-negative matrix factorization (NMF) which can capture the underlying correlation structure of genetic data, while at the same time, the positive components of output can give a better interpretation of the results.

## 2. Materials and Methods

Figure 1 shows the proposed procedure to select informative genetic variants and to identify subgroups of hypertension patients using unsupervised learning algorithms.

### 2.1. Using Logistic Regression to Filter Hypertension Related Variants

To divide the patients with hypertension into subgroups using genetic information, variants which have no association with hypertension status need to be filtered out. This is built on the assumption that variants which are correlated with the hypertension status will also function on further dividing hypertension patients into subgroups. The classical logistic regression model is constructed to reflect the relationship between the variants and hypertension status. Factors such as age, gender, and BMI are known to be related to hypertension. Therefore, we take these covariates into consideration to adjust the confounding effects.

Assume the number of subjects/patients with both genetic and phenotypic information is *n* and the number of variants in the pool (number of variants fed into the regression model) is *P*. For the *i*th subject (i=1,2,…,n), $y_{i}$ denotes his/her hypertension status (1 = with hypertension, 0 = without hypertension), $X_{i}$ = $\left(X_{i_{1}},X_{i_{2}},X_{i_{3}}\right)$ is a vector representing the three covariates, and $G_{ip}$ denotes the *p*th genetic variant for *i*the subject where p=1,2,…,P, taking values of 0, 1, or 2. The logistic model is applied to each of the *P* variants:(1)$$logit\left[P\left(y_{i}=1\right)\right]=\alpha_{0_{ip}}+\alpha_{\mathrm{ip}}^{\prime }X_{i}+\beta_{ip}G_{ip}$$

Variants with *p*-values less than a certain cutoff value are selected. We set different cutoff values to test the robustness. This basic filtration gives us the variants related to the hypertension status. However, many of them have a negative effect on hypertension status, which means that they help control the hypertension. Since our purpose was to find the informative variants who can further divide the hypertension patients into subgroups, we only considered those variants which have positive effect on hypertension status. Therefore, only variants with the positive effect are selected.

### 2.2. Using F-Statistics to Rank the Selected Variants

Single nucleotide polymorphism (SNP) has three characteristics [3]: (1) it is common in the human genome (occurs every 100–300 bases); (2) about 75% variations are from cytosine (C) to thymine (T); (3) it is stable from generation to generation. Due to these characteristics, genetic variants can be used on population classification. Variants can be ranked and selected based on certain measurements which reflect their capability of population classification. In this paper, we generalize the F Statistics which were originally used in population classification [3,8] to rank the variants selected from logistic regression. Variants with higher F statistics are more related to hypertension disease status. We add variants in decreasing order of F statistics to the unsupervised clustering model.

We assume that the SNPs which have the higher capability of distinguishing the hypertension status also play an important role in subtyping patients with hypertension. In our study, there are two sub-populations (with or without hypertension). Each feature is expressed as an SNP, and the F statistics is defined as:(2)$$F=Var\left(p\right)/\left(\bar{p}\times \bar{q}\right)$$
where $\bar{p}=\frac{p_{1}+p_{2}}{2}$, $\bar{q}=1-\bar{p}$ represent the two alleles’ mean frequencies in two population classes. $p_{1}$ and $p_{2}$ represent the frequencies of one allele for the first and the second population, respectively. $var\left(p\right)=\sum_{c=1}^{c=2}{\left(p_{c}-\bar{p}\right)}^{2}/2$.

### 2.3. Taking into Account the Impact of Rare Variants on Complex Diseases

Genome-wide association studies (GWASs) have identified over 1000 genetic loci influencing blood pressure with multiple systems and tissues implicated [9]. Common variants are widely accepted to have effect on complex disease. However, there are arguments about the effect of rare variants on explaining the trait heritability. As researchers have not derived the determined variants associated with the hypertension status, we apply the filtering process to both of rare variants and common variants. We use the gnomAD database as reference for selection of rare variants. SNPs with frequency less than 1% in the gnomAD database are selected from our dataset as rare genetic variants.

### 2.4. Aggregating Variants within the Same Genetic Region to Increase Power

Many variants in our dataset have low frequencies; therefore, the effectiveness of applying them directly to the clustering method is under powered [10,11]. A logical approach is to use the burden test which applies clustering methods to the summarizing statistics that represent the variants within the same genetic region [10,11,12,13,14,15]. Clustering and testing on the cumulative effects can increase the power.

The basic assumption for the burden test is that impact of variants within the same genetic region on the phenotype or disease status has the same direction and scale. However, this assumption can yield large bias. In our case, in the same gene, some of the variants have negative effect on hypertension status, while others have positive effect. To deal with this problem while increasing the test power, we aggregate variants with positive effect size only. These variants influence the hypertension status with the same direction and scale.

### 2.5. Selecting Deleterious Variants

Even though variants selected from logistic model are correlated with the hypertension status, they may be synonymous SNPs, which means that their change will not cause an amino acid to change. The noise brought by these synonymous SNPs may weaken the real signal. ANNOVAR gene-based annotation is used to select deleterious variants. Three databases (refGene, knownGene, ensGene) are used as references. If a variant is deleterious in any one of the three databases, then it is selected. In our case, nonsynonymous, stopgain, and stoploss variants are defined as deleterious. We use the African Americans as an example to show the number of different types of variants (please refer to Table 1).

### 2.6. Using NMF to Cluster Hypertensive Patients into Subgroups

Different unsupervised learning algorithms have been used to identify disease subtypes. The commonly used algorithms include hierarchical clustering (HC), principal component analysis (PCA), self-organizing map (SOM), etc. However, these methods have limitations when being applied to genetic data. HC has stringent tree structures and is very sensitive to the choice of dissimilarity measurement. PCA requires matrices to be orthogonal which results in difficulty of interpretation for the decomposition. SOM is unstable to the choice of initial conditions. In this paper, we use nonnegative matrix factorization (NMF) to identify subgroups of hypertension patients. As we are focusing on count data that are by definition nonnegative, we use NMF instead of other grouping methods such as k-means or principal component analysis (PCA) that do not have a built-in nonnegative constraint. NMF can also capture the underlying structure of genetic data and at the same time, it organizes both the genetic information and samples, to provide biological insight.

Given a matrix *A* of size N×M, *N* represents the number of variants (or genes in the aggregation approach), *M* represents the number of patients with hypertension. With a desired rank *K*, NMF is used to compute an approximation of A∼WH, where *W* (size N×K) and *H* (size K×M) are non-negative matrices. Each column of *W* defines the coefficients of variants in each corresponding meta-variant (or meta-gene with aggregation). H represents the level of meta-variants (or meta-genes) in each sample. We cluster samples into subgroups based on the level of meta-variants (or meta-genes). For example, sample *m*(1,2,…,M) belongs to cluster k(1,2,…,K) with the highest level of meta-variant (meta-gene) *k* in matrix *H*.

To capture the subclass, we use the “brunet” algorithm [5] which is based on iteratively updating divergence-based equations related to Poisson likelihood function. Since NMF has stochastic property, we can use the Cophenetic correlation coefficient (CCC) to determine the best rank K. The method is originally proposed by [5]. The basic assumption is that if rank K fits the data well, sample distribution should be similar under different runs. The cophenetic correlation coefficient summarizes the consistency of sample distribution under different initials. In a perfect case, the value of CCC is 1 (0≤CCC≤1). Best rank K is determined by high CCC. In this real-data analysis, CCC has the highest value when we divide the hypertension patients into two clusters both for African American and Caucasian cohorts.

### 2.7. Using Kruskal–Wallis Test to Evaluate the Efficiency of Different Methods

Nine phenotypic variables (please refer to Table 2 for detailed information) are used to examine the clinical significance of genetic-based hypertension subgroup identification. It is clinically meaningful if distributions of the phenotypic variables are significantly different among different subgroups. Since they are non-normally distributed within samples of different sizes, we apply Kruskal–Wallis test which is a non-parametric rank test used to compare two or more independent samples of equal or unequal sample sizes.

## 3. Results

We first introduce the study population. Then, we present the results for different combinations of filtration methods to select the most important and relevant variants which are used to cluster hypertensive patients into subgroups.

### 3.1. Population of Study

Our study subjects were enrolled in the Hypertension Genetic Epidemiology Network (HyperGEN), which is part of the National Institutes of Health Family Blood Pressure Program. It is a cross-sectional study consisting of individuals with hypertension, their siblings, and a random sample of normotensives who were all recruited from four cities in the United States. HyperGEN aims to identify and characterize the genetic basis of familial hypertension. Complete details of the HyperGEN study are reported in Williams et al. [16]. Both Caucasian and African American HyperGEN participants underwent GWAS with Affymetrix 5.0 and 6.0 arrays, respectively. For quality control, we removed monomorphic markers, insertion/deletion variants, and single nucleotide polymorphisms (SNPs) with missing rate >5% or Mendelian errors. For imputation, we used Minimac with 1000 G Phase I Integrated Release Version 3 Haplotypes (2010-11 data freeze, 2012-03-14 haplotypes) that contains haplotypes of 1092 individuals of all ethnic backgrounds and excludes monomorphic and singleton sites. Participants also underwent systematic echocardiography (cardiac ultrasound) as described in detail by Aguilar et al. [17]. These echocardiograms are initially recorded on VHS videotape format and subsequently digitized to DICOM format after which they underwent speckle-tracking analysis for measurement of indices of cardiac mechanics. The measurements from the HyperGEN echocardiograms that reflect cardiac mechanics are outcome variables (only nine phenotypic variables are available), as listed in Table 2. Indices of cardiac mechanics obtained by speckle-tracking echocardiography are more sensitive measures of intrinsic cardiomyocyte function [18] compared to conventional cardiac function measures such as ejection fraction. Furthermore, indices of cardiac mechanics are thought to be subclinical measures of myocardial dysfunction that occur during the transition from risk factors (e.g., hypertension, obesity, diabetes, coronary disease) to overt heart failure [19,20]. Markers of maladaptive cardiac structural remodeling such as left ventricular hypertrophy (i.e., increased left ventricular mass) can also impact indices of cardiac mechanics [17].

Of the 1258 African American HyperGEN participants with ExomeChip data, 921 had adequate echocardiograms for quantitation of indices of cardiac mechanics. Of these 921 participants, 575 had a history of hypertension. There were 2,846,152 variants in 38,189 genes or between-gene regions. Of the 1270 Caucasian HyperGEN participants, 1181 had adequate echocardiograms for quantitation of indices of cardiac mechanics. Of these 1181 participants, 612 had hypertension (2,354,081 variants in the Caucasians). Our goal is to cluster hypertensive participants using genetic information into clinically meaningful subtypes in the HyperGEN African American and Caucasian cohorts separately.

### 3.2. Comparisons of Clustering Results for Different Methods

For both African American and Caucasian cohorts, two hypertension subtypes are identified. Table 3 summarizes different combinations of filtering methods and the corresponding *p*-values from Kruskal–Wallis test for comparing distributions of phenotypic variables between two subgroups.

The first combination is designed to explore the influence of rare variants on hypertension status. The number of rare variants for African American and Caucasian are 73,738 and 4925, respectively. Of these, 1872 and 141 have positive effect size in logistic regression. We use F-statistics to rank all rare variants, choose certain cutoff numbers of variants and intersect them with the variants which have positive effect size. This step helps to reduce the false positive rate of variants selected from logistic regression model. Since the signal for rare variants is low, we aggregate variants within the same genetic region before inputting them into NMF model. For the African American cohort, the best cutoff percentage for F ranked rare variants is 2% which yields the highest clustering efficiency. The number of informative variants left is 472 covering 306 genetic regions. Six phenotypic variables are statistically different in their distributions between the two subgroups. However, there are no significant variables under different cutoff percentages for Caucasian cohort.

Since there are arguments about the influence of rare variants and common variants on complex diseases. We revise the first combination by considering all variants instead of just rare variants. The number of statistically significant variants with positive effect size from logistic regression are 83,648 and 67,450 for African American and Caucasian cohorts, respectively. Since the number of total variants sequenced is too large, we use F statistics to rank the positively effective variants and find the best cutoff percentage. Other steps are the same as for the first combination. The best clustering result is that four phenotypic variables are significantly different between two subgroups for African American patients and six for Caucasian patients. However, the number of variants used to achieve the highest clustering efficiency for two cohorts are significantly different in scale. The best cutoff percentage for African American is 15% with 12,555 variants, while for Caucasian the best cutoff is 1% with only 675 variants.

Since using F-statistics requires subjective judgment on the choice of best cutoff percentage of ranked variants and the number of variants selected is not consistent in two cohorts, we decide to apply another filtering method. First, we select deleterious variants whose changes affect the amino acids. However, not all deleterious variants are correlated with hypertension. Therefore, in the second step, we still need to build logistic model to filter out uncorrelated variants. The last four rows of Table 3 summarize the main components of this combination. To test the sensitivity of clustering efficiency, we apply different cutoff *p*-values to select significant variants from logistic model. It is noticed that the number of variants and covered genetic regions are similar, with one variant per region on average. Therefore, the combinations without gene aggregation are also conducted. In this special case, doing so will not only keep all the original genetic information but also ensures the test power. Table 3 shows that gene aggregation brings no benefit to the clustering efficiency; this may be due to the relatively high MAF of selected deleterious variants with positive effect size and the low average number of variants per genetic region. Therefore, we use the results from combinations without gene aggregation. Under different cutoff *p*-values (4th and 6th rows in Table 3), the number of statistically significant variables in African American cohort stays the same. For Caucasian cohort, six variables are significantly different in their distribution between two subgroups under 0.05 *p*-value cutoff. When we increase the cutoff *p*-value to 0.1, the number of significant variables increases to seven with six of them detected with cutoff *p*-value of 0.05. Therefore, the clustering results are robust to the choice of cutoff *p*-values. Another important thing to note is the change of cluster allocation of hypertensive patients under different cutoff *p*-values (listed in Table 4). The number of African American patients whose allocation does not change under two cutoff *p*-values is 353 out of 575 total AA hypertension patients. In Caucasian cohort, only one patient changes his/her cluster allocation. The clustering allocation of African American patients is sensitive to the choice of cutoff *p*-values, while not for Caucasian cohort. As is known, African American tend to have more genetic variants compared to Caucasian, therefore, using genetic information to distinguish the disease subtypes for African Americans is relatively challenging. We also conduct a Kruskal–Wallis test on NMF decomposition matrix H. The results show that patients whose allocation change under two different cutoff *p*-values are those with similar entries in the corresponding columns of H, which indicates these patients have ambiguous clustering allocation using NMF method.

### 3.3. Summary

Three criteria are used to select the best combination of filtering process: (1) high clustering efficiency, meaning that the number of variables whose distributions are significantly different among subgroups is high; (2) consistency between African American and Caucasian cohorts, meaning that the number of variants used for analysis and the ratio of patients’ number in each subgroup are comparable between two cohorts; (3) succinctness, meaning that the number of variants used for subtype identification is relatively low for achieving high efficiency.

Taking into consideration the above-mentioned criteria, we conclude that “logistic (cutoff *p*-value = 0.05) (positive effect size) + deleterious variants” is the best choice for our population of study.

We rank selected single nucleotide polymorphisms (SNPs) (370 SNPs for African American and 217 for Caucasian) based on their *p*-value in the logistic regression and conduct the gene set enrichment analysis for genes in which top 10 SNPs reside. For African American, genes for top 10 important SNPs are COL24A1, INSL6, MTTP, ATF7IP, CUBN, SLC39A12, ZNF208, LPIN3, UTRN; COL24A1, INSL6, MTTP, ATF7IP, CUBN, SLC39A12, and ZNF208 and are showed to have close relationship to cardiac functions [21,22,23,24,25,26]. For Caucasian cohort, genes for top 10 important SNPs are DNHD1, ASPM, TTC21B, GMPS-KCNAB1, HEATR1, OR10J1, and LGALS8; except for DNHD1, all other genes are related to cardiovascular function [27,28,29,30,31].

The number of significant variables in African American and Caucasian cohorts are two and six, respectively. The box plots of these outcome variables are presented in Figure 2 and Figure 3. Compared with other combinations, this one uses the lowest number of variants to give a consistent ratio of number of patients in two subgroups between the two cohorts. More specifically, Figure 2 suggests that genetic driven patient clusters tend to show differences in cardiac systolic functions. For example, one of the main cardiac systolic indexes, global longitudinal strain (gls), shows worse systolic mechanics in Cluster 1. Figure 3 suggests that genetic driven patient clusters tend to show differences in both cardiac systolic and diastolic functions. For example, global longitudinal strain (gls) and septal e’ velocity (eseptal), two main cardiac systolic and diastolic indexes, respectively, both show worse cardiac mechanics in Cluster 1. Interestingly, Figure 2 and Figure 3 also indicate possibly different genotype to phenotype pathways for African American and Caucasian cohorts, which calls for more follow-up studies.

## 4. Discussion

With the phenotypic variables, researchers were able to identify more than 2 hypertension subtypes such as borderline isolated systolic hypertension, definite isolated systolic hypertension, isolated diastolic hypertension and definite hypertension. Our approach is more focused on exploration on the possibility/potential to use SNP data to identify subtypes. We have expected that using SNP data, may not allow us to get much granular subtype division as using phenotypic variables, since we have much fewer patients compared to the number of SNPs. In the future research, we think two possible situations will help to increase the resolution of subtype identification: (1) get more available patients and (2) use multi-view or integrative learning to incorporate both genetic and phenotypical variables in the study. In addition, there are some limitations of the proposed approach. The chosen unsupervised learning method in our research is “brunet” NMF which assumes Poisson distributed entries of inputted matrix. The assumption holds when we aggregate variants within the same genetic-region. However, the inputted SNP array with entries 0, 1, 2 does not strictly satisfy the model assumption. Future work may focus on improving existing NMF algorithms for analysis of trinary matrix. Trinary data (i.e., values 0,1,2) have a distinctive characteristic that the features (attributes) they include have the same nature as the data they intend to account for: both are trinary. If we can use trinary matrices (W and H) to approximate the input matrix, the clustering results will be even more interpretable. The trinary matrix W explicitly designates the cluster memberships for data points and the matrix H indicates the feature representations of each cluster. The other limitation is that we have not incorporated family pedigree data as covariates in the logistic regression. Considering the computational complexity (e.g., modeling family structure as random effect for thousands of families will take weeks of computation), we have used age, gender, and BMI as covariates in the first pre-processing filtering step. In future research, we will focus on the potential to incorporate pedigree covariates into our modeling process. For other future work, we will focus on (1) considering SNPs in regards of its associated direction (i.e., positive or negative effect size) in the logistic regression; (2) systematically exploring the impact of deviations of practical and theoretical distributions for NMF application on genotyping data, e.g., testing the impact on the results if imputed SNP dosage data were used instead, with continuous variables; (3) testing our approach on an independent dataset to further verify the main findings; (4) improving our model by starting from verified blood pressure-associated SNPs.

## 5. Conclusions

In this paper, we propose a procedure that uses statistical methods to select the most informative disease-related variants and applies suitable clustering methods to identify subgroup of patients with similar genetic performance. The research gives promising results that selecting and analyzing informative genetic information can result in meaningful and clinically relevant subtypes of hypertensive patients with statistically significant difference in underlying etiology/pathophysiology and distribution of clinical outcomes. Given the heterogeneous nature of hypertension, genetic information learning may inform the development and trial of novel targeted therapies.

## Figures and Tables

**Figure 1 genes-11-01265-f001:**
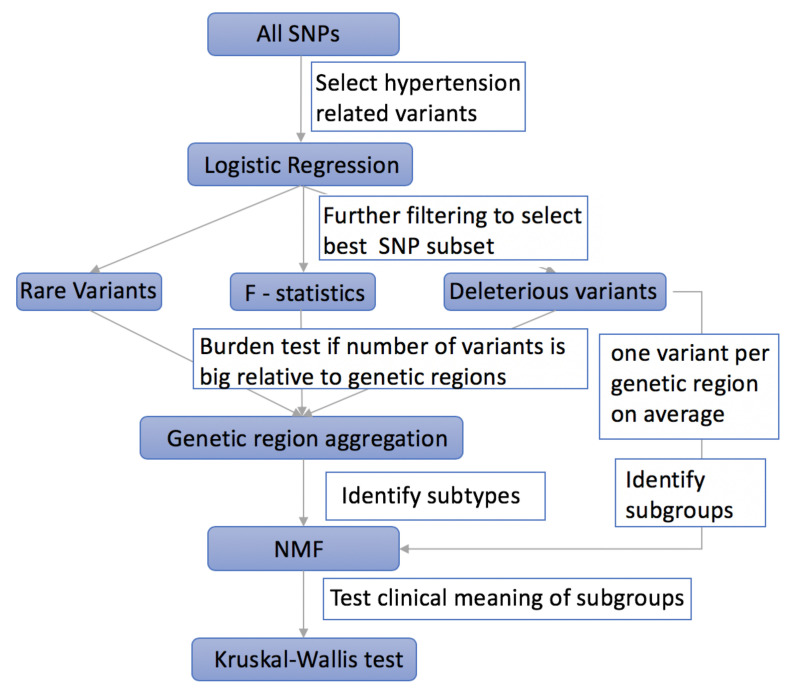
Flow chart of the procedure to select informative variants and identify hypertension subgroups.

**Figure 2 genes-11-01265-f002:**
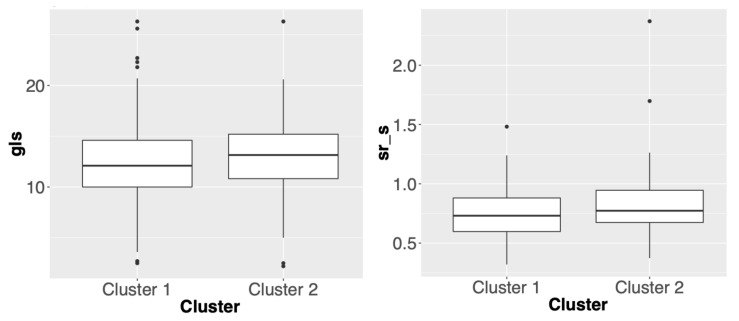
Box plots of phenotypic/outcome variables whose distributions are significantly different between two clusters in African American cohort. The *p*-value for gls is 0.0205 and the *p*-value for sr_s is 0.0141.

**Figure 3 genes-11-01265-f003:**
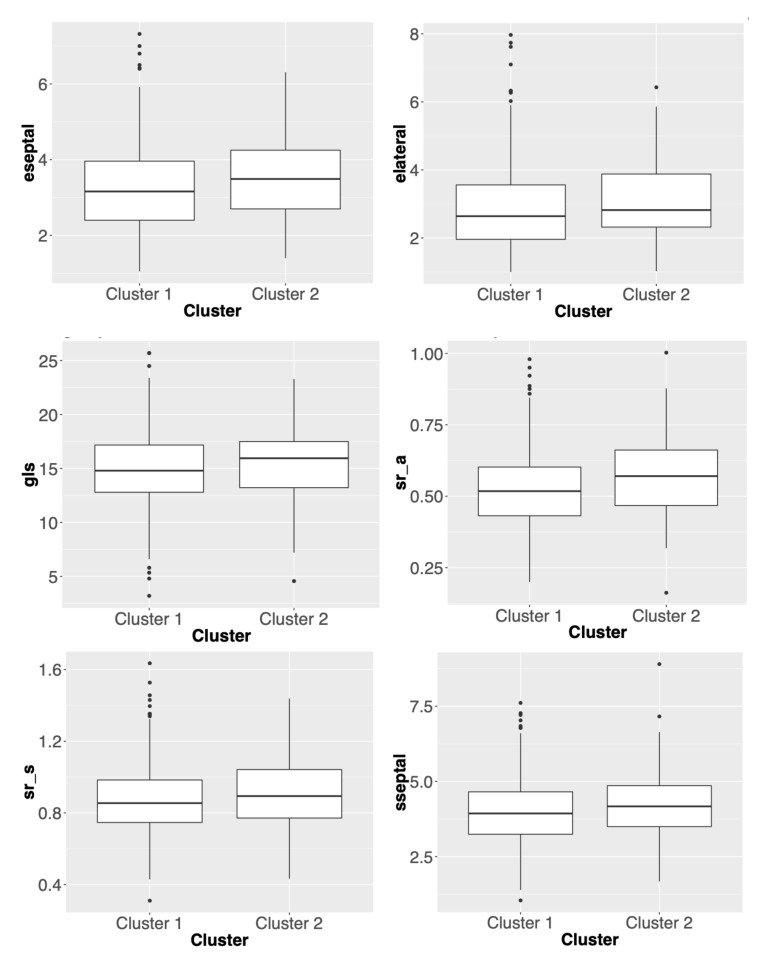
Box plots of phenotypic/outcome variables whose distributions are significantly different between two clusters in Caucasian cohort. The *p*-values for eseptal, elateral, gls, sr_a, sr_s and sseptal are 0.0135, 0.0465, 0.0246, 0.0151, 0.0774, 0.0879 respectively.

**Table 1 genes-11-01265-t001:** Summary of number of variants with different functions using three gene annotation databases for African American patients.

SNP Array	ANNOVAR Gene-Based Annotation		
AA	*refGene*	*knownGene*	*ensGene*
nonsynonymous SNP	10,554	11,073	11,404
stopgain	90	110	117
stoploss	12	17	17
synonymous SNP	17,802	18,231	18,511
unknown	553	10	9

**Table 2 genes-11-01265-t002:** Description of 9 phenotypic/outcome variables that are used to examine the significance of genetic-based hypertension subtype identification.

Abbreviation	Full Name	Description
elateral	Lateral e’ velocity	Left ventricular early diastolic relaxation velocity, measured at the lateral mitral annnulus in the apical 4-chamber view
eseptal	Septal e’ velocity	Left ventricular early diastolic relaxation velocity, measured at the septal mitral annnulus in the apical 4-chamber view
gcs	Global circumferential strain	Left ventricular circumferential strain, measured in the parasternal short axis view
gls	Global longitudinal strain	Left ventricular longitudinal strain, measured in the apical 4-chamber view
grs	Global radial strain	Left ventricular radial strain, measured in the parasternal short axis view
sr_a	strain rate-atrial	Left ventricular late (atrial) diastolic strain rate, measured in the apical 4-chamber view
sr_e	strain rate-early diastlic	Left ventricular early diastolic strain rate, measured in the apical 4-chamber view
sr_s	strain rate-early systolic	Left ventricular systolic strain rate, measured in the apical 4-chamber view
sseptal	Septal s’ velocity	Left ventricular systolic longitudinal velocity, measured at the septal mitral annulus in the apical 4-chamber view
slateral	Lateral s’ velocity	Left ventricular systolic longitudinal velocity, measured at the lateral mitral annulus in the apical 4-chamber view

**Table 3 genes-11-01265-t003:** Summary of Kruskal–Wallis test results for each phenotypic variable using different combination of filtrations.

Methods *p*-Value ${}^{a}$ (AA/Caucasian)	Phenotypic Variables								
	Elateral	Eseptal	gcs	gls	grs	sr_a	sr_e	sr_s	Sseptal
Logistic (0.05) + F + Rare ${}^{b}$ + geneaggr ${}^{c}$	0.11 />0.1	**0.07** />0.1	0.97 />0.1	**0.02** />0.1	0.80 />0.1	**0.07** />0.1	**0.06** />0.1	**0.01** />0.1	**0.00** />0.1
Logistic (0.05) + F + All ${}^{d}$ + geneaggr	0.51 /**0.04**	0.32 /**0.00**	**0.05** /**0.03**	**0.07** /**0.01**	0.18 /0.28	**0.04** /0.44	0.11 /**0.05**	**0.08** /0.18	0.16 /**0.02**
Logistic (0.05) + del ${}^{e}$ + geneaggr	0.97 /**0.04**	0.68 /**0.01**	0.73 /0.65	**0.01** /**0.03**	0.76 /0.39	0.19 /**0.02**	0.62 /0.10	**0.01** /**0.08**	0.11 /**0.09**
Logistic (0.05) + del	0.76 /**0.05**	0.30 /**0.01**	0.42 /0.65	**0.02** /**0.03**	0.74 /0.39	0.48 /**0.02**	0.64 /0.10	**0.01** /**0.08**	0.29 /**0.09**
Logistic (0.1) + del + geneaggr	0.49 /**0.07**	0.11 /**0.02**	0.97 /0.63	0.16 /**0.03**	0.72 /0.37	0.13 /**0.01**	0.29 /0.14	0.30 /**0.09**	0.39 /0.11
Logistic (0.1) + del	0.77 /**0.03**	0.82 /**0.02**	0.89 /0.73	**0.02** /**0.02**	0.65 /0.32	0.21 /**0.02**	0.45 /**0.08**	**0.02** /**0.06**	0.15 /**0.06**

[a] The first row shows the name of 9 phenotypic variables; [b] “Rare” means only rare variants are used; [c] “geneaggr” means aggregating variants within the same genetic region; [d] “All” means all variants are used; [e] “del” means deleterious variants.

**Table 4 genes-11-01265-t004:** The first half is the summary of number of significant phenotypic variables and number of variants selected to do the clustering analysis using different combinations of filtrations. The second half summarizes the number of hypertensive patients in each subgroup under different combinations of filtrations.

Methods/Number of Significant Phenotypic Variables	Number of Significant Phenotypic Variables (Number of Variants; Number of Genetic Regions If with Geneagg)		Number of Hypertensive Patients in Each Cluster	
	African American	Caucasian	African American	Caucasian
Logistic (0.05) + F + Rare + geneaggr	6 (472)	0		
Logistic (0.05) + F + All + geneaggr	4 (12,555)	6 (675)	430/145	583/29
Logistic (0.05) + del + geneaggr	2 (370; 339)	6 (217; 213)	318/257	485/127
Logistic (0.05) + del	2 (370)	6 (217)	445/130	485/127
Logistic (0.1) + del + geneaggr	0 (735, 643)	6 (467, 379)	398/177	486/126
Logistic (0.1) + del	2 (735)	7 (467)	273/302	484/128

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
