# Peer review of "Genetic-Based Hypertension Subtype Identification Using Informative SNPs"

_genes, 2020, doi:10.3390/genes11111265_

Round 1

Reviewer 1 Report

The authors have developed a statistical process for using genetic information to cluster hypertensive individuals into different subtypes of hypertension. Their method firstly uses logistic regression to select a set of informative SNPs associated with a positive risk of hypertension, then uses F-statistics to rank the SNPs, followed by an unsupervised machine learning algorithm (NMF) to cluster the individuals according to these SNPs. The method is applied to both a Caucasian and an African American cohort of the HyperGen study. By testing the two different produced clusters for association with nine cardiac mechanic phenotypes from echocardiogram data, the authors conclude that the two different clusters correspond to clinically meaningful subtypes of hypertension.

The authors are addressing a hot topic, for genetic clustering of disease and phenotype subtyping. I therefore found it interesting to read. This is novel in the literature for hypertension. They have chosen their methodology carefully, and explain the reasons for their choices clearly within the text.

Some of the methodology may be statistically technical for the average reader, although Figure 6 is a very helpful addition to the paper, to provide a good overview of the whole pipeline, so I would strongly advise that this figure remains in the final version as a main figure.

However the main weaknesses of this study are: (i) that only one method is applied, with no comparison of other similar approaches; (ii) there is no validation of the method or the resulting findings in an independent cohort; (iii) the choice of the nine phenotypes for providing evidence of clinical meaning are very narrow/specific.

I therefore think that whilst this paper would be of interest to many readers, and makes a very good attempt, it is important to acknowledge that it is only the start of a big goal, contributing only one initial step. Nevertheless, I believe publication of this work would help to inspire others in the field, encourage and aid future larger efforts.

Major comments:

  1. As noted above, there is no validation of the method or the resulting findings in an independent cohort. Due to the novelty, this study would benefit from validation, especially within both of these AA and Caucasian ancestries, as well as within other ancestries, especially in light of the comments in lines 165-167.
  2.  
  3. Is it statistically valid to perform all stages of this process on the same data? For example, should the logistic regression have been performed on a separate independent training subset of the data?
  4. I am surprised that the authors did not begin by using the published set of SNPs known to be associated with blood pressure from BP-GWAS. Instead, they used the same data to run their own logistic regression genetic analysis for hypertension. (a) Please could the authors explain why? (b) It would also be helpful and interesting to the reader if the authors provided a full supplementary table of their logistic regression SNP results. (c) And furthermore if their top associated SNPs in their dataset could be compared to the published BP-associated SNPs. For example, not many of the genes listed in lines 150-154 seem to be well-established BP-associated genes. (d) Linking to my comment above, use of published genetic variants could remove the need for the initial logistic regression analysis stage, and mean that the selection of the SNPs was independent from an external dataset.
  5. It seems strange that the authors only consider the nine echocardiogram phenotypes listed in Fig 1. I would assume that the HyperGen study contains many more available cardiovascular related phenotypes of interest than this. I would be interested in further analyses comparing the final two different cluster subtypes with other cardiovascular risk factor phenotype traits and outcomes. This would further improve the testing for clinical meaning and relevance of the cluster outputs, and suggest more direction for clinical translation.
  6. Lines 77-82 are the only place where the initial genetic data is described, so lots of important information is lacking: (a) Which ExomeChip exactly does the data come from? I am surprised that ExomeChip data contains as many as 2,486,152 variants, so assuming this must be a recent version of an array like the OmniExome, for example… (b) Please provide a basic summary of all the quality control applied to the genetic data used. (c) Could the analysis be improved by using imputed genetic data instead? In particular, noting comments in lines 171-173, regarding the assumption of the models, in relation to the 0, 1, 2 SNP data values…how would the methodology be impacted if imputed SNP dosage data were used instead, with continuous variables?
  7. The methodology and research design used is quite specific. In particular, only one specific approach is performed. Could an alternative method be applied to this same HyperGen data, and the cluster results be compared?
  8. Figure 2, especially, and also Figure 3 are not at all easy to follow and interpret. There is too much detailed information crammed into one Table. In addition to the boxplots in Figures 4 & 5 (which are useful, but not very visually exciting), I wonder whether any of the cluster results and other data summarised in Figures 2 & 3 could be presented more graphically?
  9. The p-value cut-offs of P<0.05 or P<0.1 for the logistic regression analysis seem very liberal. Please could the authors justify this? For example, this is a lot more liberal than any standard GWAS analysis would be, for identifying hypertension-associated genetic variants.
  10. In Figure 5, I note that two of the significantly claimed phenotypes have p-value > 0.05. Please could the authors therefore clearly state, and justify, their p-value threshold for this final stage of the analysis? This extra detail is also required in the text of Section 4.7. In particular, has multiple testing been accounted for, with regards to testing nine different phenotypes? In which case, please comment on whether these nine phenotypes are either phenotypically or genetically correlated with each other?

Minor comments:

  1. The abstract gives the false impression that the results are based on 1,258 AA and 1,270 Caucasian individuals. However, in lines 77-82, we realise that only 575 AA and 612 Caucasian hypertensives are actually included in the main analysis. Please state the N more appropriately in the abstract, and make the N very clear in the text within each stage of the analysis process.
  2. The statement “GWAS have identified more than 1,000 genetic loci associated with human diseases and traits” in lines 219-220 is extremely out of date! In fact it is from a 2009 reference! Please could the authors update this information? Indeed, there are actually now over 1,000 genetic loci published for association with blood pressure or hypertension itself!
  3. Within the discussion, I would be interested for the authors to discuss how the findings compared to their initial expectations. For example, did the authors expect the clustering to only result in two distinct subtype clusters? Or were they expecting there to be more? Particularly knowing from the initial background discussion in the Introduction, that many different subtypes of hypertension are believed to exist. Furthermore, how would the authors expect to be able to identify additional subtypes of hypertension in future research?
  4. The use of genetic risk scores is very popular in the field at the moment. As further analysis, I wonder whether the authors could construct GRS from the selected SNPs in the model, and compare the distributions of GRS between the different clusters, for example?
  5. The authors discuss how this genetic analysis is more challenging in the AA cohort than the Caucasians, and in several places, highlight many differences in the results between the two ancestries. I am therefore surprised that their criteria (2) in Summary Section 2.3 is consistency between the AA and Caucasian cohorts. Considering all the other differences, do the authors still expect the datasets to behave similarly enough, in order for this to be a performance evaluation criteria?
  6. In lines 197-199 the covariates age, gender and BMI are stated. (a) Should the top principal components from the genetic data also be used? (b) Also, please confirm if there is any relatedness or family structure within the HyperGen study which would need to be taken into account?
  7. The notation is often confusing or not ideal. For example, in Section 4.6, the notation “N” refers to the number of variants, whereas in most literature this is much more commonly used to denote the number of samples / individuals.
  8. I was interested to discover in lines 200-201 that only variants with a positive risk effect on hypertension were selected. I can see later in Section 4.4 that this helped with the Burden test assumptions. However, for bi-allelic SNPs, couldn’t all SNPs be considered, and the SNP alleles be aligned such that the risk-increasing allele was the reference direction for the effect of the SNP in the subsequent models? Or were you also taking into account whether the risk allele was the major or minor allele…?
  9. In lines 150-154, many of the genes listed are repeated, which seems confusing…?
  10. Please check line 272, where the figure label is missing with “??”.

Reviewer 2 Report

A promising scientific research is about assessing the significance of racial differences in genetic determination of cardiovascular system structural changes in arterial hypertension patients. 

Some questions: what kind of classification did you use to detect the diagnosis of arterial hypertension? (American or European)

There were some mistakes in the section of Introduction: lines 24 & 25 - the correct border level of arterial pressure to detect the arterial hypertension is ≥ 140 but not ≥ 160 as I think as cardiologist.

Author Response

  1. Some questions: what kind of classification did you use to detect the diagnosis of arterial hypertension? (American or European)

Thank you. It's American.

  1. There were some mistakes in the section of Introduction: lines 24 & 25 - the correct border level of arterial pressure to detect the arterial hypertension is ≥ 140 but not ≥ 160 as I think as cardiologist.

Thank you. The data we are showing in line 24&25 are quoted from the reference “R.A.K.E.E.V. Gupta, and A.K. Sharma, “Prevalence of hypertension and subtypes in an Indian rural population: clinical and electrocardiographic correlates,” Journal of Human hypertension, 8(11), pp.823-829, Nov 1994.”

Round 2

Reviewer 1 Report

Thank you for a few of the changes that you did make to the manuscript, in response to my reviewer comments.

I still have a few remaining minor edits to suggest to the authors, please, in response to your rebuttal. (I will use the original numbering, for convenience)

Previous major comments:

(1) If future validation in an independent study is agreed to be a good idea for the future, then please could this be added as a comment in the future work section of the Discussion.

(3) I still think these two suggestions would be quick to do and helpful for the reader and the overall field, to be included in the paper, please:

(b) It would also be helpful and interesting to the reader if the authors provided a full supplementary table of their logistic regression SNP results.

(c) And furthermore, if their top associated SNPs in their dataset could be compared to the published BP-associated SNPs. For example, not many of the genes listed in lines 150-154 seem to be well-established BP-associated genes

Also, replying further to your response here, then depending on how you define "verified", I don't think I agree with this comment: "Although there are published set of SNPs known to be associated with blood pressure, but the number of verified SNPs is small"...as I don't think over 1,000 identified genetic loci from BP-GWAS is at all a small number of SNPs!

(4) OK, then please state this fact clearly in the text: "For the HyperGEN dataset in our research, only these 9 phenotypic variables are available."

(6) Some of these comments about other methods could be mentioned in Discussion perhaps then, please.

(7) I'm afraid that I still think the Tables in Figs 2 & 3 are very busy, and difficult for a reader to interpret, so hence are going to be difficult for a reader to make good use of, once printed in journal. Perhaps discuss with the journal editor if there is space to make them bigger and lay them out better...?

(8) With regards to having used p-value cut-offs of 0.05 or 0.1...In basic statistics, yes. But not in statistical genetics, so I disagree. What about adjustment for multiple testing, as you are testing many thousands of SNPs…? What would the Bonferroni-level of significance be adjusted for multiple testing according to the number of pairwise independent variants being tested?

(9) Ok, but then please make sure that all p-value thresholds for all parts of the analysis are clearly stated and justified in the text.

Previous minor comments:

(3) Ok, please add some comments like this in the Discussion then

(6) I'm not sure if I agree with this statement from a statistical genetics theoretical/methodology perspective: "Although HyperGEN does have family pedigree data, our reasoning is that top principal components and family structure will not have a cohort wide impact on SNP-hypertension association".

So I think just include as a comment in the limitations, please, to acknowledge that you have made this 1st step as simple as possible. e.g. like your comment in the rebuttal here: "Also considering the computational complexity (e.g., modeling family structure as random effect for thousands of families will take weeks of computation), we have used age, gender and BMI as covariates in the first pre-processing filtering step."

(8) Yes, I understand about the Burden test. But, I still don't think you've replied to my comment here: "However, for bi-allelic SNPs, couldn’t all SNPs be considered, and the SNP alleles be aligned such that the risk-increasing allele was the reference direction for the effect of the SNP in the subsequent models?"

In contrast to your reply here: "The other reason is that we think only SNPs that can contribute to the hypertension have the power to distinguish the hypertension subtypes.", I would still say: "Yes, but one allele of the SNP will be risk for HTN, and the other protective for HTN…so all SNPs could still all be used if they have a significant association with HTN, and then just be aligned to the risk allele…"

Author Response

Previous major comments:

(1) If future validation in an independent study is agreed to be a good idea for the future, then please could this be added as a comment in the future work section of the Discussion.

Thank you for the suggestion. We have added this to the discussion.

(3) I still think these two suggestions would be quick to do and helpful for the reader and the overall field, to be included in the paper, please:

(b) It would also be helpful and interesting to the reader if the authors provided a full supplementary table of their logistic regression SNP results.

Thank you for the suggestion. We have provided the logistic regression SNP results as supplementary tables.

(c) And furthermore, if their top associated SNPs in their dataset could be compared to the published BP-associated SNPs. For example, not many of the genes listed in lines 150-154 seem to be well-established BP-associated genes

Also, replying further to your response here, then depending on how you define "verified", I don't think I agree with this comment: "Although there are published set of SNPs known to be associated with blood pressure, but the number of verified SNPs is small"...as I don't think over 1,000 identified genetic loci from BP-GWAS is at all a small number of SNPs!

Thank you very much for the suggestion. We agree that it’s a good idea to (1) starting from known associated BP-associated SNPs (2) verify the top associated SNPs in the model with known BP-associated SNPs. We have put it in the discussion part as the future work focus. In the current work, we have compared the top genes in our model with some BP-associated genes (line 150-154). These genes are listed from BP-associated research papers, we do think that they should be well-established BP-associated genes.

(4) OK, then please state this fact clearly in the text: "For the HyperGEN dataset in our research, only these 9 phenotypic variables are available."

Thanks for the comment. We have added it explicitly in the main paper.

(6) Some of these comments about other methods could be mentioned in Discussion perhaps then, please.

Thank you for the suggestion. We have added the comments about other competitor clustering methods in section 4.6.

(7) I'm afraid that I still think the Tables in Figs 2 & 3 are very busy, and difficult for a reader to interpret, so hence are going to be difficult for a reader to make good use of, once printed in journal. Perhaps discuss with the journal editor if there is space to make them bigger and lay them out better...?

Thank you for the suggestion. We will further discuss with the editor to find a better way to present these two figures.

(8) With regards to having used p-value cut-offs of 0.05 or 0.1...In basic statistics, yes. But not in statistical genetics, so I disagree. What about adjustment for multiple testing, as you are testing many thousands of SNPs…? What would the Bonferroni-level of significance be adjusted for multiple testing according to the number of pairwise independent variants being tested?

Thank you for the comment. (1). The basic idea of our approach is similar to the sure independence screening (SIS, proposed by Fan and Lv, 2008) which reduces the ultrahigh-dimensional covariates to moderate dimensional covariates by filtering out those that are irrelevant covariates, using a marginal screening procedure. In SIS, it’s not necessary to perform the multiple-testing correction and under certain assumptions, the probability of selecting right covariates is approaching to 1. Even though, our application does not satisfy all the model assumptions, it’s reasonable to use it as a proxy approach.  (2) Logistic regression is just the first filtering process; it’s followed by other filtering approaches later. Therefore, we would not recommend using Bonferroni correction. Since Bonferroni is too conservative which might lead to severe false negatives.

(9) Ok, but then please make sure that all p-value thresholds for all parts of the analysis are clearly stated and justified in the text.

Thank you for the suggestion. We have checked the description in main text and made sure we have stated it clearly.

Previous minor comments:

(3) Ok, please add some comments like this in the Discussion then.

Thank you for the suggestion. We have added the comments to Section 3 discussion in the main text.

(6) I'm not sure if I agree with this statement from a statistical genetics theoretical/methodology perspective: "Although HyperGEN does have family pedigree data, our reasoning is that top principal components and family structure will not have a cohort wide impact on SNP-hypertension association".